materials science

poly(ethylene imine), triethylenetetramine, chitosan microspheres, adsorption, dye wastewater

**Author for correspondence:**
Jianlan Cui
e-mail: 414559203@qq.com

This article has been edited by the Royal Society of Chemistry, including the commissioning, peer review process and editorial aspects up to the point of acceptance.

# Preparation of aminated chitosan microspheres by one-pot method and their adsorption properties for dye wastewater

Jian Meng, Jianlan Cui, Siyuan Yu, Hui Jiang, Congshan Zhong and Ji Hongshun

School of Chemical Engineering and Technology, North University of China, Taiyuan 030051, People's Republic of China

(ID) SY, 0000-0002-7557-5234

Polyamine chelating adsorbents have a good removal effect on dye wastewater. In this study, small molecule triethylenetetramine and macromolecular poly(ethylene imine) were selected as aminated reagent, and two kinds of aminated chitosan microspheres, TETA-CTSms and PEI-CTSms, were obtained by emulsion cross-linking method. The microspheres were fully characterized by FTIR, SEM, XRD, EDS and TGA. EDS results showed that the N content of the PEI-CTSms and TETA-CTSms microspheres increased significantly after the cross-linking reaction and can reach 5.7 wt% and 7.3 wt%, respectively. Adsorption experiments confirmed that TETA-CTSms and PEI-CTSms showed greater adsorption characteristics for anionic dye reactive yellow (RY) in aqueous solutions compared with CTSms, and the adsorption capacity per unit area was increased by 39.11% and 88.56%, respectively. The adsorption capacity of aminated microspheres for RY decreased with the increase of pH. The adsorption kinetics conformed to the pseudo-second-order model, and the adsorption process was in accordance with the Langmuir isotherm model. The negative value of $\Delta G$ confirmed that the adsorption process was spontaneous, and the dye adsorption was a multiple process dominated by chemical chelating and physical adsorption.

## 1. Introduction

Dye wastewater has become a serious problem that restricts rapid economic development, affects people's health and seriously damages the environment [1–3]. Most dyes are toxic organic

**Scheme 1.** Chemical process of preparing aminated chitosan microspheres.

compounds of strong chemical stability, they are hard to degrade and have carcinogenic, teratogenic and mutagenic effects. These poisonous compounds seeping into underground aquifers from the surface contaminate groundwater, causing a direct impact on drinking water. Therefore, it is very urgent to develop new technologies and materials for removing pollutants from water [4,5].

Polyamine chelating adsorbent has wide application prospects in the field of wastewater treatment. According to the soft hard acid base principles, N atoms, as the intermediate hard alkali, are more easily bound with intermediate hard acids to form stable complexes [6]. Polyamine chelating adsorbents have many advantages, such as good removal effect, strong selectivity, high efficiency of separation and recovery, reliable operation and so on. Chitosan is mainly a linear polysaccharide composed of glucopyranose units. It is a derivative obtained by deacetylation of chitin. It has a wide range of sources, low cost and good biocompatibility, and is widely used in the food, pharmaceutical and medical industries [7,8]. The presence of amine groups and hydroxyl groups on the chitosan molecule can be used as an excellent biosorbent for the treatment of industrial wastewater [9–12].

It is reported that chitosan-based materials have been used in different forms. Sheet and powder materials are not suitable for use as adsorbents due to their small surface area, low mechanical strength and low porosity [13]. The preparation of beads or microspheres by emulsifying cross-linking can effectively solve the above problems [14,15], and the commonly used cross-linkers are glutaraldehyde, epichlorohydrin, ethylene glycol diglycidyl ether (EGDE), etc. [16]. However, the decrease of the adsorption site is attributed by the cross-linking reaction, especially in the case of reaction with amine groups [17].

To improve the adsorption performance, it has been reported that adsorbent material was modified by high chelating coordination of S and N functional groups [18–21]. However, the reaction steps to achieve polyamine design were numerous in these reports. Jia *et al.* [18] prepared triethylenetetramine aminated chitosan derivatives used epichlorohydrine as a medium through two reactions. Elwakeel [19] used glutaraldehyde and epichlorohydrine as a medium to bond triethylenetetramine with the chitosan chain through three reactions. Jing *et al.* [20] prepared poly(ethylene imine) aminated chitosan derivatives with the aid of cross-linking agent glutaraldehyde by two reactions. In order to reduce the amount of solvents and reagents used in the multi-step experiment, in this work, aminated chitosan materials were prepared by simultaneous emulsion cross-linking of polyamine reagents. Specific reaction steps are shown in scheme 1.

# 2. Experimental

## 2.1. Reagents and instruments

Glutaraldehyde (GA, 50% in water), chitosan (CTS, deacetylation degree ≥90%), hydrochloric acid (HCl, 36–38%), triethylenetetramine (TETA), methylene blue (MB), sodium hydroxide and sodium chloride were purchased from Sinopharm Chemical Reagent Co. Ltd; poly(ethylene imine) (PEI, molecular weight of $2 \times 10^4$–$5 \times 10^4$) was purchased from Wuhan Qianglong Co. Ltd; reactive yellow (RY) was purchased from Shanghai Jiaying Chemical Co. Ltd; deionized water was made in the laboratory.

Infrared spectrometer (FTIR, Spectrum Two, USA); laser particle size distribution analyser (BT-2002, Dandong Bettersize); scanning electronic microscope (SEM, SU8010, Japan); X-ray diffraction (XRD, Dandong Haoyuan); UV–visible spectrophotometer (UV, UV-2602, China) were used as analytical instruments.

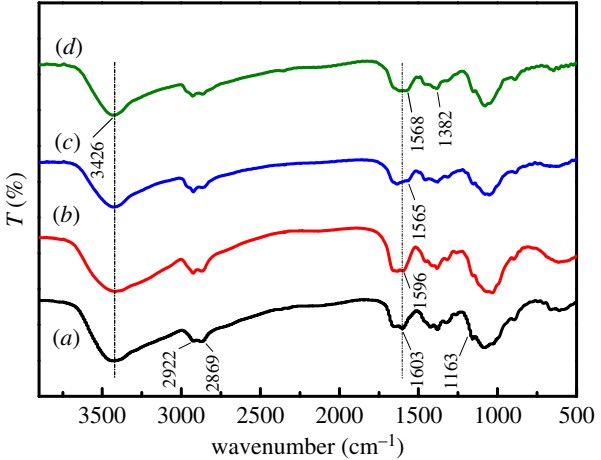

**Figure 1.** IR spectra of four particles (*a*) CTS powder, (*b*) CTSms, (*c*) TETA-CTSms, (*d*) PEI-CTSms.

## 2.2. Preparation of aminated chitosan microspheres

First, 100 ml of 2% chitosan-acetic acid solution was configured; 0.25 ml TETA was dissolved in 5 ml water, and then 1 ml glacial acetic acid was used to adjust the solution to weak acidity. The above-mentioned solutions were poured together, and mixed by ultrasonic mixing. TETA-CTSms chitosan microspheres were prepared by using micro emulsion cross-linking technology, in which 50 ml TETA-chitosan solution, 100 ml liquid paraffin, 5 ml Span-80 and 2 ml 25% glutaraldehyde were used as water phase, oil phase, emulsifier and cross-linking agent respectively.

Meanwhile, the preparation of the PEI-CTSms microspheres was similar to the above steps. The structure of the aminated chitosan microsphere was confirmed by FTIR, SEM, TGA, XRD and EDS.

## 2.3. Study on adsorption properties of TETA-CTSms and PEI-CTSms

MB and RY were used as adsorption object in the adsorption experiment; the concentration of them in the solution was determined by using UV analysis, and the pH value of dye solution was adjusted by HCl and NaOH aqueous solution.

The adsorption kinetics were carried out by using TETA-CTSms and PEI-CTSms as adsorbents; 4 mg chitosan microspheres were introduced into 40 ml dye solution with different concentration. The isothermal adsorption experiments were performed in a constant temperature oscillator. After adsorption saturation, the concentration of MB or RY in the supernatant was determined by using UV. The equilibrium adsorption amount was calculated according to equation (2.1).

$$Q_e = \frac{V(C_0 - C_e)}{m},\qquad(2.1)$$

where $Q_e$ (mg g$^{-1}$) is the equilibrium adsorption amount; $V$ (ml) is the volume of the adsorption liquid or eluent, respectively; $C_0$ and $C_e$ (mg l$^{-1}$) are the initial and equilibrium concentration of dye solution, respectively; $m$ is the quality of adsorbents.

# 3. Results and discussion

## 3.1. Characterization of chitosan microspheres

Figure 1 shows the infrared spectra of CTS powder, CTSms, TETA-CTSms and PEI-CTSms.

In the spectrum of chitosan powder (figure 1*a*), the broad peak near 3426 cm$^{-1}$ is formed by the overlap of -OH and -NH$_2$ stretching vibrations. Two absorption peaks can be observed at 2922 cm$^{-1}$ and 2869 cm$^{-1}$ in the infrared spectrum which can be attributed to the antisymmetric and symmetric stretching vibrations of -CH$_2$. The bending vibration of the amine group in the chitosan generates an absorption peak at wavelength of 1603 cm$^{-1}$. In the spectrum of three types of chitosan microspheres, the absorption peaks of amine at 1603 cm$^{-1}$ are shifted. The corresponding absorption peak of CTSms is shifted to 1596 cm$^{-1}$ (figure 1*b*), and that of the two aminated modified microspheres,

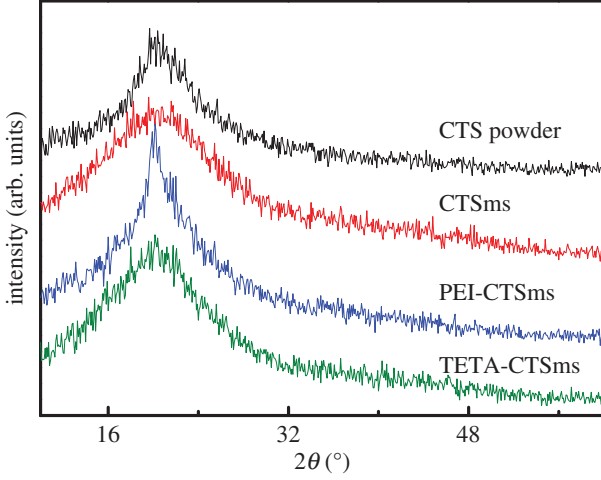

**Figure 2.** XRD patterns of CTS powder, CTSms, PEI-CTSms and TETA-CTSms.

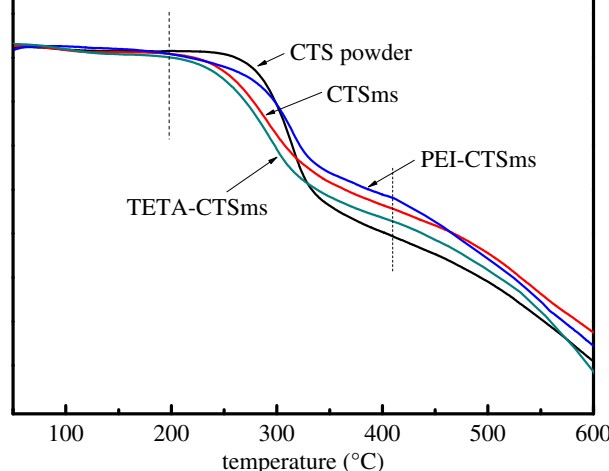

**Figure 3.** TG curves of CTS powder, CTSms, PEI-CTSms and TETA-CTSms.

TETA-CTSms and PEI-CTSms, are shifted to 1565 and 1568 cm$^{-1}$ (figure 1$c$,$d$), respectively. These show that the amine participates in the reaction during the cross-linking reaction [22]. In addition, it can also be observed that the characteristic peak at 3426 cm$^{-1}$ becomes sharper. This may be due to the co-cross-linking with the aminated substance TETA and PEI, and the content of the amine increases in the obtained microspheres, resulting in the enhancement of infrared absorption.

As shown in figure 2, compared with the chitosan powder, it can be observed that the diffraction peak intensity of CTSms at 20.14° becomes weak. This may be due to the fact that the regularity of the chitosan chain is reduced by the cross-linking reaction, resulting in a decrease in the crystallinity [23]. In contrast, the diffraction peak intensity of TETA-CTSms and PEI-CTSms at 20.14° is enhanced. This may be because, in the process of simultaneous co-cross-linking with aminated substance, a large number of hydrogen bonds can be formed with the chitosan chains, thereby leading to the increase in the crystallinity.

In addition, figure 3 shows the TG curves of various particles. Due to the regular arrangement of the chitosan chain being destroyed in the cross-linking reaction, the initial degradation temperature of the three cross-linking microspheres is lower than that of CTS powder.

Figure 4 shows the SEM images of CTSms (*a*), TETA-CTSms (*b*) and PEI-CTSms (*c*). As shown in figure 4, the sphericity of CTSms microspheres is regular, the surface of microspheres is smooth, and the size distribution of microspheres is relatively concentrated. The degree of cross-linking reaction increases during the preparation of aminated microspheres, resulting in poor dispersion of TETA-CTSms and PEI-CTSms microspheres.

In addition, the N content on the surface of the three kinds of microspheres was analysed by EDS. The N content of the PEI-CTSms and TETA-CTSms microspheres increased significantly after the cross-linking reaction (figure 5).

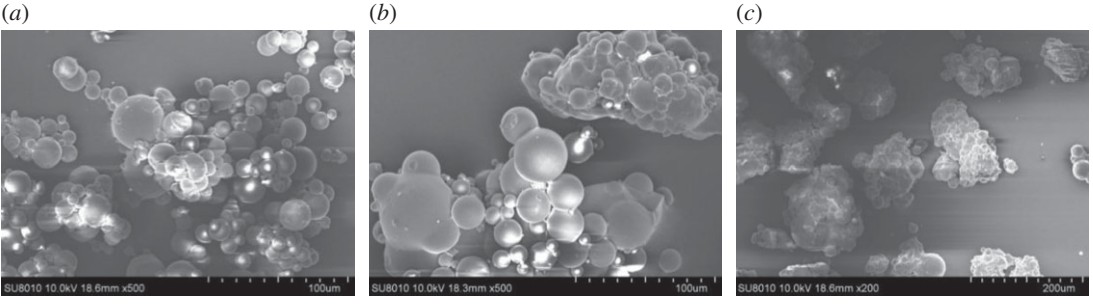

**Figure 4.** SEM images of (*a*) CTSms, (*b*) PEI-CTSms and (*c*) TETA-CTSms.

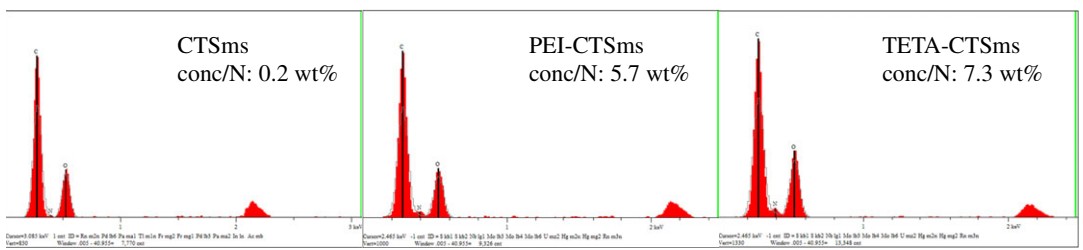

**Figure 5.** EDS for CTSms, PEI-CTSms and TETA-CTSms.

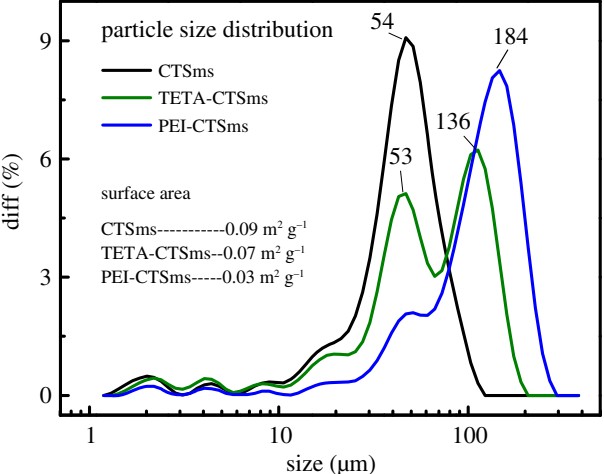

**Figure 6.** Particle size distribution for CTSms, PEI-CTSms and TETA-CTSms.

As shown in figure 6, the particle size distribution of CTSms and PEI-CTSms microspheres is relatively narrow, concentrating on 54 and 184 µm, respectively, while the diameter distribution of TETA-CTSms microspheres is relatively broad. In addition, the surface areas of CTSms, TETA-CTSms and PEI-CTSms microspheres are 0.09, 0.07 and 0.03 $m^2 g^{-1}$, respectively. The surface area of PEI-CTSms is the lowest because the microspheres are coated together.

## 3.2. Study on adsorption properties of microspheres

### 3.2.1. Study on adsorption kinetics

Figure 7 illustrates the effect of contact time on the adsorption of MB and RY using CTSms, TETA-CTSms and PEI-CTSms microspheres. Furthermore, the pseudo-first-order kinetic model and

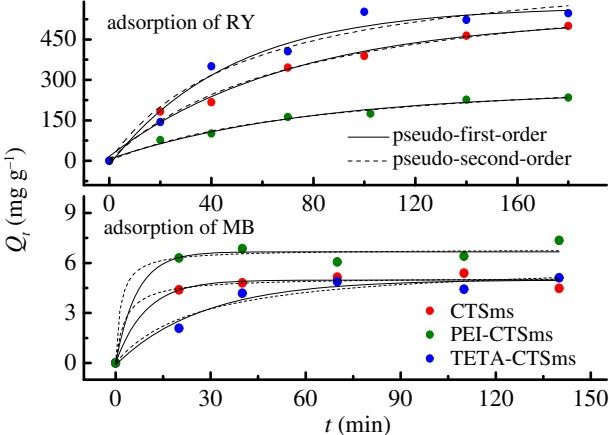

**Figure 7.** Effect of adsorption time on adsorption capacity of microspheres for MB and RY; pseudo-first/second order kinetic models.

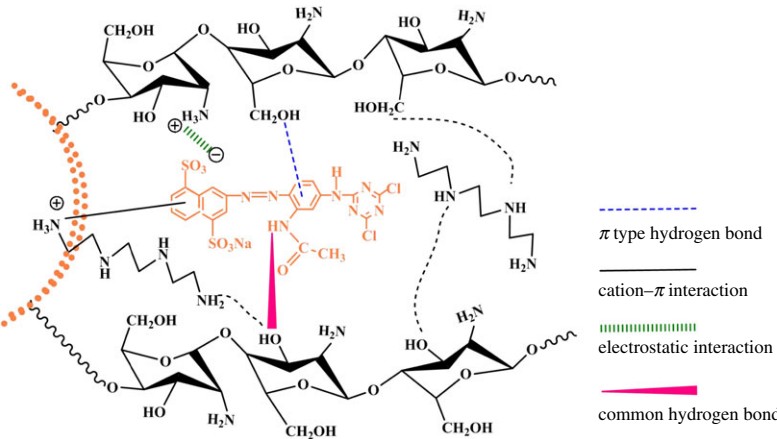

**Scheme 2.** Schematic expression of the interaction between TETA-CTSms microspheres and RY.

pseudo-second-order kinetic model were adopted to fit the kinetic adsorption data, and the corresponding formulae are as follows [24]:

$$\text{pseudo-first-order rate equation:}\quad \ln(Q_e - Q_t) = \ln Q_e - k_1 t$$

$$\text{pseudo-second-order rate equation:}\quad \frac{t}{Q_t} = \frac{1}{k_2 Q_e^2} + \frac{t}{Q_e},$$

where $Q_t$ and $Q_e$ are the adsorption capacities of microspheres for dye at time $t$ and adsorption equilibrium, respectively; $k_1$ and $k_2$ are equilibrium adsorption rate constants.

As shown in figure 7, the adsorption capacities of TETA-CTSms and PEI-CTSms for MB are 4.72 and 6.77 mg g$^{-1}$ in 2 h and that for RY is 513.61 and 199.07 mg g$^{-1}$. According to scheme 2, there should be mainly four interaction forces between aminated chitosan microspheres and RY molecules, including hydrogen bonding, π-type hydrogen bond, cation–π interaction and electrostatic interaction. Particularly, the aminated microspheres contain a large number of amino groups which are mostly protonated, resulting in strong electrostatic adsorption of anionic dye RY. In contrast, its adsorption to cationic dyes MB is weak.

As shown in table 1, in the adsorption process of three microspheres for MB and RY, the value of the linear correlation coefficient $R^2$ obtained by pseudo-second-order kinetic is large, and the equilibrium adsorption capacity $Q_{exp}$ obtained by fitting is close to the experimental value, indicating that the adsorption is mainly controlled by chemisorption [25]. Combined with the surface area of three materials, the adsorption capacities of CTSms, TETA-CTSms and PEI-CTSms for MB are 63.44, 120.00 and 270.33 mg m$^{-2}$, and that for RY are 7891.56, 10977.14 and 14879.33 mg m$^{-2}$, respectively. The adsorption effects of the TETA-CTSms and PEI-CTSms aminated microspheres are significantly improved.

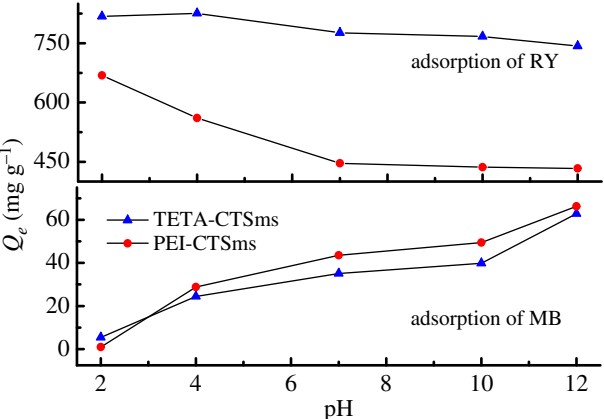

**Figure 8.** Effect of pH on adsorption capacity of MB and RY.

**Table 1.** Adsorption kinetic constants for chitosan microspheres.

| | $Q_{exp}$ (mg g$^{-1}$) | pseudo-first-order kinetic model | | | pseudo-second-order kinetic model | | |
|---|---|---|---|---|---|---|---|
| | | $Q_e$ (mg g$^{-1}$) | $K_1$ (min$^{-1}$) | $R^2$ | $Q_e$ (mg g$^{-1}$) | $K_2$ (g mg$^{-1}$ min$^{-1}$) | $R^2$ |
| adsorption of MB | | | | | | | |
| CTSms | 5.71 | 4.98 | 0.105 | 0.962 | 5.13 | 0.066 | 0.967 |
| PEI-CTSms | 8.40 | 6.69 | 0.148 | 0.957 | 6.81 | 0.092 | 0.969 |
| TETA-CTSms | 8.11 | 5.11 | 0.036 | 0.943 | 6.08 | 0.006 | 0.937 |
| adsorption of RY | | | | | | | |
| CTSms | 710.24 | 535.14 | 0.014 | 0.978 | 701.23 | $1.91 \times 10^{-5}$ | 0.986 |
| TETA-CTSms | 768.40 | 571.41 | 0.021 | 0.961 | 749.05 | $2.45 \times 10^{-5}$ | 0.956 |
| PEI-CTSms | 446.38 | 259.70 | 0.013 | 0.977 | 349.32 | $3.30 \times 10^{-5}$ | 0.984 |

### 3.2.2. Effect of pH and ionic concentration on adsorption

Figure 8 shows the adsorption changes of TETA-CTSms and PEI-CTSms for MB and RY with different pH values. It can be observed that the adsorption capacity of TETA-CTSms and PEI-CTSms microspheres for MB increases with the increase of pH. When the pH is 12, the adsorption amount reaches 62.86 and 66.22 mg g$^{-1}$, respectively. In the acidic condition, the amino group on the microspheres is in protonation state, so the adsorption capacity is low due to the electrostatic repulsion to MB with positive charge [19]. But the electrostatic repulsion weakens with the increase of pH, making the adsorption quantity increase. Since the RY is negatively charged in aqueous solution, the adsorption result is reversed. When the pH is 2, the adsorption amount reaches 818.06 and 668.76 mg g$^{-1}$, respectively.

It can be seen from figure 9 that the adsorption capacity of TETA-CTSms for MB and RY decreases with the increase of NaCl concentration in solution. Meanwhile, the adsorption capacity of PEI-CTSms for MB decreases with the increase of NaCl concentration, while that for RY is just the opposite. The reason may be that as the concentration of NaCl increases, the solubility of MB and RY in solution decreases slightly, which leads to its continuous diffusion to the solid–liquid interface, resulting in the increase of corresponding adsorption capacity. In addition, a large amount of Cl$^-$ and Na$^+$ in the solution will compete with the dye for adsorption, and weaken the adsorption between the adsorbents, eventually leading to a decrease in the amount of adsorption [23,26]. These two factors lead to the adsorption results shown in figure 9.

### 3.2.3 Isothermal adsorption study and parameter fitting

As shown in figure 10, the adsorption quantity of TETA-CTSms for RY increases with the increase of initial concentration of RY. In addition, the adsorption capacity of TETA-CTSms for RY increases with the increase of temperature, indicating that adsorption is an endothermic process. In addition,

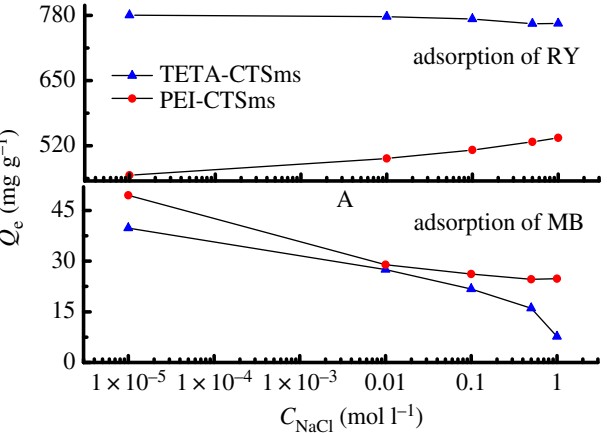

**Figure 9.** Effect of ionic concentration on adsorption capacity of MB and RY.

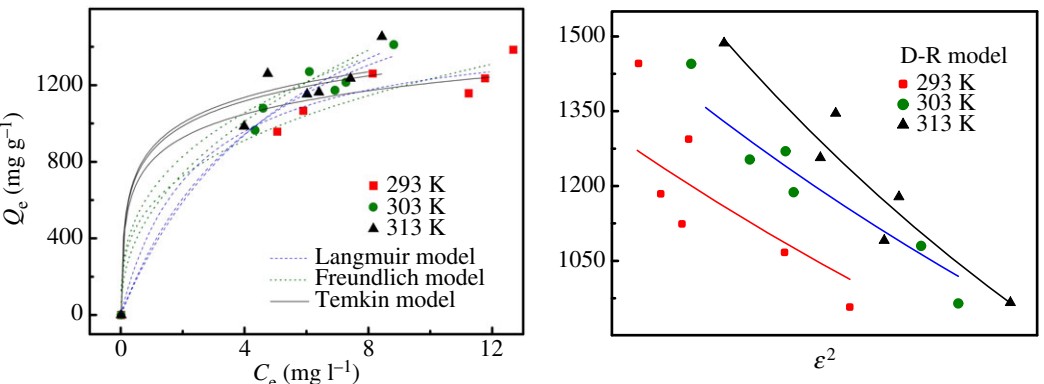

**Figure 10.** Adsorption isotherms of RY adsorption by TETA-CTSms at different temperatures, Langmuir, Freundlich, D-R isotherm and Tempkin models of RY adsorption on TETA-CTSms.

different isothermal adsorption models were used to fit the above isothermal adsorption data, and the corresponding formulae are as follows [27–29]:

$$\text{Langmuir isothermal equation:} \quad \frac{C_e}{Q_e} = \frac{1}{b_L Q_m} + \frac{C_e}{Q_m}$$

$$b_L = (K - 1) \times \frac{M}{\rho}$$

$$R_L = \frac{1}{(bC_0 + 1)}$$

$$\text{Freundlich isothermal equation:} \quad \ln Q_e = \ln K_F + \frac{\ln C_e}{n}$$

$$\text{D-R isothermal equation:} \quad \ln Q_e = \ln Q_m - K_D \varepsilon^2$$

$$\varepsilon = RT \ln\left(1 + \frac{1}{C_e}\right)$$

$$E = \frac{1}{\sqrt{-2K_D}}$$

$$\text{Temkin isothermal equation:} \quad Q_e = \frac{RT}{b_T} \ln(K_T C_e).$$

Table 2 lists the fitting parameters obtained from the four isothermal adsorption models. The Langmuir isothermal adsorption model has the best fitting effect, followed by the Temkin isothermal adsorption model. It shows that the adsorption of TETA-CTSms for RY is mainly monolayer adsorption. The values of the dimensionless parameter $R_L$ are all between 0 and 1, which means that the conditions are favourable for adsorption. The $1/n$ values of Freundlich fitting parameters are

**Table 2.** Parameters of isotherm models of RY adsorption by TETA-CTSms.

| model | parameter | $T$ (°C) | | |
|---|---|---|---|---|
| | | 20 | 30 | 40 |
| Langmuir | $Q_m$ (mg g$^{-1}$) | 1518.63 | 2111.03 | 2327.25 |
| | $b_L$ (l mg$^{-1}$) | 0.212 | 0.289 | 0.329 |
| | $R_L$ | 0.015 – 0.023 | 0.031 – 0.047 | 0.037 – 0.056 |
| | $K$ | 1.012 | 1.016 | 1.018 |
| | $R^2$ | 0.913 | 0.987 | 0.985 |
| Freundlich | $K_F$ (l mg$^{-1}$) | 576.45 | 539.27 | 647.98 |
| | $1/n$ | 0.330 | 0.453 | 0.352 |
| | $R^2$ | 0.883 | 0.916 | 0.848 |
| D-R | $Q_m$ (mg g$^{-1}$) | 1345.34 | 1557.15 | 1848.58 |
| | $K_D$ | 1.465 | 1.555 | 2.092 |
| | $R^2$ | 0.647 | 0.735 | 0.884 |
| Temkin | $b_T$ (J mol$^{-1}$) | 13.79 | 13.11 | 13.69 |
| | $K_T$ | 94.72 | 92.87 | 98.96 |
| | $R^2$ | 0.904 | 0.898 | 0.868 |

**Table 3.** Thermodynamics parameters at different temperatures.

| $T$ (°C) | $\Delta G$ (kJ mol$^{-1}$) | $\Delta H$ (kJ mol$^{-1}$) | $\Delta S$ (J K$^{-1}$ mol$^{-1}$) |
|---|---|---|---|
| 20 | − 28.58 | 244.76 | 0.935 |
| 30 | − 39.99 | | |
| 40 | − 47.19 | | |

between 0 and 1 at different temperatures, indicating that the adsorption process is preferential adsorption [30]. In addition, according to the equilibrium constant $K$ at different temperature, the Gibbs free energy $\Delta G$ was obtained. The values of $\Delta S$ and $\Delta H$ were calculated by using the Van't Hoff equation [31]. The thermodynamic data of the adsorption process are shown in table 3.

$$\Delta G = -RT \ln K$$

and

$$\ln K = -\frac{\Delta H}{RT} + \frac{\Delta S}{R}.$$

As shown in table 3, the $\Delta G$ values at different temperatures are negative, which indicates that the adsorption process is spontaneous. Both $\Delta H$ and $\Delta S$ are positive. The results show that the adsorption process is endothermic reaction, and the degree of freedom increases in the solid–liquid interface during the adsorption process. Judging by the $\Delta H$ value, the adsorption process of RY onto TETA-CTSms can be defined as chemical adsorption [31].

Compared with most adsorbents for the adsorption of RY presented in table 4, the aminated microspheres exhibited superior adsorption performance. Various factors including adsorption conditions, surface structure and functional groups will affect the adsorption capacities jointly. Also, for dye molecules with similar charge groups and molecular size, aminated microspheres showed a larger adsorption capacity.

**Table 4.** Adsorption capacities of various adsorbents.

| adsorbent | | temperature (°C) | solution pH | $Q_{max}$ (mg g$^{-1}$) | ref. |
|---|---|---|---|---|---|
| cellulose-based adsorbent | cellulose | ambient | 7.0 | 6.31 | [32] |
| | hPEI modified | | | 90.50 | |
| cucurbiturils mixtures | CB[6] | 25 | | 289.1 | [33] |
| | CB[8] | | | 2135.4 | |
| chitosan 8B | | 30 | 4.0 | 133.25 | [34] |
| chitosan composite CTS/MMT | | 50 | 3.0 | 317.23 | [35] |
| composite fungal biomasses | | 25 | 5.0 | 87.60 | [36] |
| polyurethane-immobilized biosorbent | | 25 | 2.0 | 116.5 | [37] |
| activity mesoporous carbons | AC-AS | 30 | 1.0 | 1397.4 | [38] |
| | AC-CP | | | 1428.4 | |
| modified mesoporous silica Py-MS | | 25 | 3.5 | 60.0 | [39] |
| aminated microspheres | TETA-CTSms | 20 | 7.0 | *768.40* | this work |
| | PEI-CTSms | | | *446.38* | |

# 4. Conclusion

In this work, PEI and TETA were synchronous emulsion co-cross-linked with chitosan to prepare PEI-CTSms and TETA-CTSms aminated microspheres. Characterization of PEI-CTSms and TETA-CTSms using various techniques confirmed its successful fabrication. The adsorption tests indicated PEI-CTSms and TETA-CTSms are effective adsorbents for the adsorption of anionic dye RY from aqueous solutions. The adsorption capacity of TETA-CTSms and PEI-CTSms microspheres for RY decreased with the increase of pH. Furthermore, the adsorption kinetics and isotherms of RY on PEI-CTSms and TETA-CTSms followed the pseudo-second-order model and the Langmuir model, respectively. The negative value of ΔG confirmed that the adsorption process was spontaneous. Based on the results of thermodynamics and effects of pH and ionic strength, it was concluded that the dye adsorption was a multiple process dominated by chemical chelating and physical adsorption.

Data accessibility. All of the data in this investigation have been reported in the paper and are freely available, and the electronic supplementary materials are also provided.

Authors' contributions. J.C., S.Y. and J.M. participated in all procedures including the design of the study. H.J., C.Z. and J.H. made substantial contributions to the acquisition and analysis of data. J.M. and S.Y. drafted and revised the article. J.C. gave final approval for publication and agreed to be accountable for all aspects of the work.

Competing interests. The authors have no competing interests.

Funding. This work was supported by Fund for Postgraduate of North University of China (project no. 20151230).

Acknowledgements. We thank Yang Xiang and Baoyun Ye for their assistance with the data collection.

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
