## [Reviewer comments · Royal Society Open Science]

Review History

RSOS-181923.R0 (Original submission)

Review form: Reviewer 1

Is the manuscript scientifically sound in its present form?

No

Are the interpretations and conclusions justified by the results?

No

Is the language acceptable?

No

Is it clear how to access all supporting data?

Yes

Do you have any ethical concerns with this paper?

No

Have you any concerns about statistical analyses in this paper?

No

Recommendation?

Reject

Comments to the Author(s)

1. There are several, but very important issues, have not been present in the manuscript. Like "the disperse property of the MSs", "the surface area (BET) infoamation", "the recycle properties of the dye adsorbent"...

2. SEM in Fig 4 show those MSs were seriously agglomerated as they been produced. The agglomerated adsorbents present perfect dye remove performance. The conclusion is seemed contradictory.

3. Abstract, should contain some important quantitative findings

Introduction, first paragraph, need to put these references:

Carbohydrate polymers 113, 115-130 (2014).

Journal of hazardous materials 177 (1), 70-80 (2010).

Royal Society open science, 4(9), 170697 (2017)

Review form: Reviewer 2

Is the manuscript scientifically sound in its present form?

Yes

Are the interpretations and conclusions justified by the results?

No

Is the language acceptable?

Yes

Is it clear how to access all supporting data?

Yes

Do you have any ethical concerns with this paper?

No

Have you any concerns about statistical analyses in this paper?

No

Recommendation?

Major revision is needed (please make suggestions in comments)

Comments to the Author(s)

In the research the authors were studied the preparation of aminated chitosan microspheres by one-pot method and their adsorption properties for dye wastewater. They investigated Characterization of chitosan microspheres with the analysis such as FT-IR, XRD, SEM, TG and EDS, and they also investigated adsorption kinetic constants for chitosan microspheres, thermodynamics parameters and the effects of parameters such as pH on the adsorption process.

1. More quantitative measurements be mentioned in the abstract

2. Summary of the section on the conclusion is also presented in the abstract
3. In the introduction section used more literature. The following references should be cited in this section for upgrading the quality of the manuscript:
 - 3.1. Removal of reactive blue 29 dye by adsorption on modified chitosan in the presence of hydrogen peroxide, *Environment Protection Engineering*, 42(1), pp. 149-168.
 - 3.2. Synthesis of nanochitosan for the removal of fluoride from aqueous solutions: A study of isotherms, kinetics, and thermodynamics, *Fluoride*, 50(2), pp. 256-268.
 - 3.3. Synthesis and performance evaluation of chitosan prepared from Persian gulf shrimp shell in removal of reactive blue 29 dye from aqueous solution (Isotherm, thermodynamic and kinetic study), *Iranian Journal of Chemistry and Chemical Engineering*, 36(3), pp. 25-36.
 - 3.4. Optimization of humic acid removal by adsorption onto bentonite and montmorillonite nanoparticles. *Journal of Molecular Liquids*, 259, pp. 76-81
 - 3.5. Investigation of equilibrium, kinetics and thermodynamics of extracted chitin from shrimp shell in reactive blue 29 (RB-29) removal from aqueous solutions. *Desalination and Water Treatment*, 70, pp. 355-363.
 - 3.6. Equilibrium and kinetics study of reactive dyes removal from aqueous solutions by bentonite nanoparticles. *Desalination and Water Treatment*, 97, pp. 329-377
4. For characterization of adsorbents, because the process was adsorption, so the analysis BET must be made to determine the surface area of the particles.
5. Formulas related to calculation of kinetics and thermodynamics in the section of experimental expressed not in the results section.
6. The conclusion is poorly written and must be rewritten.

Decision letter (RSOS-181923.R0)

11-Dec-2018

Dear Mr Yu:

Manuscript ID: RSOS-181923

Title: "Preparation of aminated chitosan microspheres by one-pot method and their adsorption properties for dye wastewater"

Thank you for submitting the above manuscript to Royal Society Open Science. Your paper was sent to reviewers and their comments are included at the bottom of this letter.

In view of the concerns raised by the reviewers, the manuscript has been rejected in its current form. However, a new manuscript may be submitted which takes into consideration these comments.

Please note that resubmitting your manuscript does not guarantee eventual acceptance, and that your resubmission will be subject to peer review before a decision is made.

Once you have revised your manuscript, go to <https://mc.manuscriptcentral.com/rsos> and login to your Author Center. Click on "Manuscripts with Decisions," and then click on "Create a

Resubmission" located next to the manuscript number. Then, follow the steps for resubmitting your manuscript.

Your resubmitted manuscript should be submitted by 10-Jun-2019. If you are unable to submit by this date please contact the Editorial Office.

On behalf of the Subject Editor Professor Anthony Stace and the Associate Editor Dr Ya-Wen Wang

REVIEWER(S) REPORTS:

Associate Editor Comments to Author ():

RSC Associate Editor:

Comments to the Author:

(There are no comments.)

RSC Subject Editor:

Comments to the Author:

(There are no comments.)

Reviewers' Comments to Author:

Reviewer: 1

Comments to the Author(s)

1. There are several, but very important issues, have not been present in the manuscript. Like "the disperse property of the MSs", "the surface area (BET) infoamation", "the recycle properties of the dye adsorbent"...
 2. SEM in Fig 4 show those MSs were seriously agglomerated as they been produced. The agglomerated adsorbents present perfect dye remove performance. The conclusion is seemed contradictory.
 3. Abstract, should contain some important quantitative findings
- Introduction, first paragraph, need to put these references:
Carbohydrate polymers 113, 115-130 (2014).
Journal of hazardous materials 177 (1), 70-80 (2010).
Royal Society open science, 4(9), 170697 (2017)

Reviewer: 2

Comments to the Author(s)

In the research the authors were studied the preparation of aminated chitosan microspheres by one-pot method and their adsorption properties for dye wastewater. They investigated Characterization of chitosan microspheres with the analysis such as FT-IR, XRD, SEM, TG and EDS, and they also investigated adsorption kinetic constants for chitosan microspheres, thermodynamics parameters and the effects of parameters such as pH on the adsorption process.

1. More quantitative measurements be mentioned in the abstract
2. Summary of the section on the conclusion is also presented in the abstract
3. In the introduction section used more literature. The following references should be cited in this section for upgrading the quality of the manuscript:
 - 3.1. Removal of reactive blue 29 dye by adsorption on modified chitosan in the presence of hydrogen peroxide, *Environment Protection Engineering*, 42(1), pp. 149-168.
 - 3.2. Synthesis of nanochitosan for the removal of fluoride from aqueous solutions: A study of isotherms, kinetics, and thermodynamics, *Fluoride*, 50(2), pp. 256-268.
 - 3.3. Synthesis and performance evaluation of chitosan prepared from Persian gulf shrimp shell in removal of reactive blue 29 dye from aqueous solution (Isotherm, thermodynamic and kinetic study), *Iranian Journal of Chemistry and Chemical Engineering*, 36(3), pp. 25-36.
 - 3.4. Optimization of humic acid removal by adsorption onto bentonite and montmorillonite nanoparticles. *Journal of Molecular Liquids*, 259, pp. 76-81
 - 3.5. Investigation of equilibrium, kinetics and thermodynamics of extracted chitin from shrimp shell in reactive blue 29 (RB-29) removal from aqueous solutions. *Desalination and Water Treatment*, 70, pp. 355-363.
 - 3.6. Equilibrium and kinetics study of reactive dyes removal from aqueous solutions by bentonite nanoparticles. *Desalination and Water Treatment*, 97, pp. 329-377
4. For characterization of adsorbents, because the process was adsorption, so the analysis BET must be made to determine the surface area of the particles.
5. Formulas related to calculation of kinetics and thermodynamics in the section of experimental expressed not in the results section.
6. The conclusion is poorly written and must be rewritten.

Author's Response to Decision Letter for (RSOS-181923.R0)

See Appendix A.

RSOS-182226.R0

Review form: Reviewer 1

Is the manuscript scientifically sound in its present form?

No

Are the interpretations and conclusions justified by the results?

No

Is the language acceptable?

Yes

Is it clear how to access all supporting data?

Not Applicable

Do you have any ethical concerns with this paper?

No

Have you any concerns about statistical analyses in this paper?

No

Recommendation?

Major revision is needed (please make suggestions in comments)

Comments to the Author(s)

1. BET should give data and graphs according to the style in the literature, not screen shots. Ref (1) Chem. Mater. 2001, 13, 3169-3183; (2) Journal of Industrial and Engineering Chemistry, Volume 21, 25 January 2015, Pages 369-377; (3) Colloid Polym Sci (2018) 296:59-70

2. My foregoing question of "SEM in Fig 4 show those MSs were seriously agglomerated as they been produced. The agglomerated adsorbents present perfect dye remove performance. The conclusion is seemed contradictory."

Author's reply: During the preparation of the aminated microspheres, due to the addition of the polyamine-based substances TETA and PEI, the degree of cross-linking between the molecular chains was increased, causing aggregation. But the amino group content on the same surface area was significantly increased, which in turn led to a higher adsorption amount.

Thus, please provide evidence data about "the amino group content on the same surface area was significantly increased". This claimed conclusion should be proved by Solid data, not description.

Review form: Reviewer 2

Is the manuscript scientifically sound in its present form?

Yes

Are the interpretations and conclusions justified by the results?

Yes

Is the language acceptable?

Yes

Is it clear how to access all supporting data?

Yes

Do you have any ethical concerns with this paper?

No

Have you any concerns about statistical analyses in this paper?

No

Recommendation?

Accept as is

Comments to the Author(s)

Now the manuscript can be accepted for publication.

Decision letter (RSOS-182226.R0)

21-Jan-2019

Dear Mr Yu:

Title: Preparation of aminated chitosan microspheres by one-pot method and their adsorption properties for dye wastewater
Manuscript ID: RSOS-182226

The editor assigned to your paper has now received comments from reviewers. We would like you to revise your paper in accordance with the referee and Subject Editor suggestions which can be found below (not including confidential reports to the Editor). Please note this decision does not guarantee eventual acceptance.

Please submit a copy of your revised paper before 13-Feb-2019. Please note that the revision deadline will expire at 00.00am on this date. If we do not hear from you within this time then it will be assumed that the paper has been withdrawn. In exceptional circumstances, extensions may be possible if agreed with the Editorial Office in advance. We do not allow multiple rounds of revision so we urge you to make every effort to fully address all of the comments at this stage. If deemed necessary by the Editors, your manuscript will be sent back to one or more of the original reviewers for assessment. If the original reviewers are not available we may invite new reviewers.

Please also include the following statements alongside the other end statements. As we cannot

publish your manuscript without these end statements included, if you feel that a given heading is not relevant to your paper, please nevertheless include the heading and explicitly state that it is not relevant to your work.

• Ethics statement

Please clarify whether you received ethical approval from a local ethics committee to carry out your study. If so please include details of this, including the name of the committee that gave consent in a Research Ethics section after your main text. Please also clarify whether you received informed consent for the participants to participate in the study and state this in your Research Ethics section.

OR

Please clarify whether you obtained the necessary licences and approvals from your institutional animal ethics committee before conducting your research. Please provide details of these licences and approvals in an Animal Ethics section after your main text.

OR

Please clarify whether you obtained the appropriate permissions and licences to conduct the fieldwork detailed in your study. Please provide details of these in your methods section.

RSC Associate Editor
Comments to the Author:
(There are no comments.)

Reviewers' Comments to Author:
Reviewer: 2

Comments to the Author(s)
Now the manuscript can be accepted for publication.

Reviewer: 1

Comments to the Author(s)

1. BET should give data and graphs according to the style in the literature, not screen shots. Ref (1) Chem. Mater. 2001, 13, 3169-3183; (2) Journal of Industrial and Engineering Chemistry, Volume 21, 25 January 2015, Pages 369-377; (3) Colloid Polym Sci (2018) 296:59–70
2. My foregoing question of "SEM in Fig 4 show those MSs were seriously agglomerated as they been produced. The agglomerated adsorbents present perfect dye remove performance. The conclusion is seemed contradictory."

Author's reply: During the preparation of the aminated microspheres, due to the addition of the polyamine-based substances TETA and PEI, the degree of cross-linking between the molecular chains was increased, causing aggregation. But the amino group content on the same surface area was significantly increased, which in turn led to a higher adsorption amount.

Thus, please provide evidence data about "the amino group content on the same surface area was significantly increased". This claimed conclusion should be proved by Solid data, not description.

Author's Response to Decision Letter for (RSOS-182226.R0)

See Appendix B.

Decision letter (RSOS-182226.R1)

25-Jan-2019

Dear Mr Yu:

Title: Preparation of aminated chitosan microspheres by one-pot method and their adsorption properties for dye wastewater
Manuscript ID: RSOS-182226.R1

It is a pleasure to accept your manuscript in its current form for publication in Royal Society Open Science. The chemistry content of Royal Society Open Science is published in collaboration with the Royal Society of Chemistry.

RSC Associate Editor
Comments to the Author:
(There are no comments.)

Reviewer(s)' Comments to Author:

Appendix A

Dear Editor,

Thank you so much for your time and suggestion on our manuscript (RSOS-181923). We have revised the manuscript thoroughly as you and the reviewers suggested. Each concern raised by the reviewers has been seriously considered and addressed in our response as below. All the significant changes have been highlighted in colors in the revised manuscript. We would greatly appreciate it if you could approve the publication of the revised manuscript. If you have any more questions or advice, please let us know.

We clarify that the screen shoots of Excel spreadsheets are not previously reported. Per the instructions for authors, the data supporting the conclusions drawn in our manuscript has been provided via electronic supplementary materials according to your requirement.

Yours sincerely,

Prof. Jianlan Cui

Reviewer #1: Comments to the Author(s)

Author's reply: We highly appreciate the reviewer's time and constructive suggestions on our manuscript. All the corrections have been made following the reviewer's suggestions and highlighted in RED in the revised manuscript.

1. There are several, but very important issues, have not been present in the manuscript. Like "the disperse property of the MSs", "the surface area (BET) information", "the recycle properties of the dye adsorbent".

Author's reply: The SEM images (Fig. 4) showed that the dispersion of the microspheres TETA-CTSms and PEI-CTSms was worse than that of the microspheres CTSms. Meanwhile, the particle size distribution curve of aminated microspheres was presented in Fig. 6. This may be because during the crosslinking of the polyamine substance, the degree of crosslinking increased due to a large amount of amino groups, and the dispersibility of the microspheres deteriorated.

The surface area of the MSs was characterized by N₂ adsorption/desorption isotherms at 77 K on Accelerated Surface Area and Porosimetry System (ASAP 2020M, Micromeritics Company), and the surface area obtained by the test was almost absent. This may be due to the tight entanglement between the molecular chains of chitosan during the process of crosslinking and precipitation when adding NaOH solution. The BET test data of CTSms, TETA-CTSms and PEI-CTSms microspheres were given.

Sample	Operator	Submitter	File	Started	Completed	Report Time	Sample Mass	Cold Free Space	Ambient Temperature	Automatic Degas
MJ-CTSms	dnb		D:\2020\DATA\2018\4\MJ-CTSms.SMP	7/28/2018 8	7/28/2018 1	8/5/2018 1	0.1023 g	84.5240 cm	22.00 °C	No
TETA-CTSms	dnb		D:\2020\DATA\2018\4\TETA-CTSms.SMP	7/28/2018	7/28/2018	8/5/2018	0.1023 g	84.5240 cm	22.00 °C	No
PEI-CTSms	dnb		D:\2020\DATA\2018\4\PEI-CTSms.SMP	7/28/2018	7/28/2018	8/5/2018	0.1023 g	84.5240 cm	22.00 °C	No

Sample	Relative Quantity	Absolute Quantity	Elapsed Time	Saturation
MJ-CTSms - Adsorption	0.058579	60.35839	-0.03373	01:15
Relative Quantity Adsorbed (nmol/g)	0.09063	93.38409	-0.05768	01:17
	0.118435	122.0332	-0.0783	01:20
	0.146352	150.7988	-0.10073	01:22

	A	B	C	D	E	F	G	H	I	J	K	L	M	N
1														
2	ASAP 2020 V4.03 (V4.03	Unit 1	Serial #: 2183	Page 1		ASAP 2020 Unit 1	Serial #: Page 1					ASAP 2020 Unit 1	Serial #: Page	
3														
4	Sample:	MJ-PEI-CTSms				Sample:	MJ-PEI-CTSms				Sample:	MJ-PEI-CTSms		
5	Operator:	dnh				Operator:	dnh				Operator:	dnh		
6	Submitter:					Submitter:					Submitter:			
7	File:	D:\2020\DATA\2018\4\MJ-PEI-CTSms.SMP				File:	D:\2020\DATA\2018\4\MJ-PEI-CTSms.SMP				File:	D:\2020\DATA\2018\4\M		
8														
9														
10														
11	Started:	7/28/2018 Analysis Adsorptive N2				Started:	7/28/2018 Analysis N2				Started:	7/28/2018 Analysis N2		
12	Completed:	7/28/2018 Analysis Bath Temp. -195.709 露				Completed:	7/28/2018 Analysis -195.709 露				Completed:	7/28/2018 Analysis -19		
13	Report Time:	8/5/2018 Thermal Correction: No				Report T18/5/2018 Thermal C No					Report T18/5/2018 Thermal C No			
14	Sample Mass:	0.1071 g Warm Free Space: 27.0121 cm ³ Measure				Sample M#0.1071 g Warm Free 27.0121 cm ³ Measure					Sample M#0.1071 g Warm Free 27.	0		
15	Cold Free Space:	82.9712 cm ³ Equilibration Inter: 10 s				Cold Free 82.9712 cm ³ Equilibr: 10 s					Cold Free 82.9712 cm ³ Equilibr: 10 s			
16	Ambient Temperature:	22.00 露 Low Pressure Dose: None				Ambient 22.00 露 Low Press: None					Ambient 22.00 露 Low Press: Non			
17	Automatic Degas:	Yes				Automatic: Yes					Automatic: Yes			
18														
19														
20														
21														
22														
23														
24														
25	Summary Report					Isotherm Tabular Report					Isotherm Linear Plot			
26														
27														
28														
29	No summary reports could be produced.													
30														
31														
32														

	A	B	C	D	E	F	G	H	I	J	K	L	M	N	O
1															
2	ASAP 2020 V4.03 (V4.03	Unit 1	Serial #: 2183	Page 1		ASAP 2020 Unit 1	Serial #: Page 1					ASAP 2020 Unit 1	Serial #: Page 1		
3															
4	Sample:	MJ-TETA-CTSms				Sample:	MJ-TETA-CTSms				Sample:	MJ-TETA-CTSms			
5	Operator:	dnh				Operator:	dnh				Operator:	dnh			
6	Submitter:					Submitter:					Submitter:				
7	File:	D:\2020\DATA\2018\4\MJ-TETA-CTSms.SMP				File:	D:\2020\DATA\2018\4\MJ-TETA-CTSms.SMP				File:	D:\2020\DATA\2018\4\MJ-TET			
8															
9															
10															
11	Started:	7/29/2018 Analysis Adsorptive: N2				Started:	7/29/2018 Analysis N2				Started:	7/29/2018 Analysis N2			
12	Completed:	7/29/2018 Analysis Bath Temp. : -195.710				Completed:	7/29/2018 Analysis -195.710 露				Completed:	7/29/2018 Analysis -195.71			
13	Report Time:	8/5/2018 Thermal Correction: No				Report T18/5/2018 Thermal C No					Report T18/5/2018 Thermal C No				
14	Sample Mass:	0.0881 g Warm Free Space: 26.9156 cm ³ Measure				Sample M#0.0881 g Warm Free 26.9156 cm ³ Measure					Sample M#0.0881 g Warm Free 26.9156	0			
15	Cold Free Space:	83.6708 cm ³ Equilibration Interval: 10 s				Cold Free 83.6708 cm ³ Equilibr: 10 s					Cold Free 83.6708 cm ³ Equilibr: 10 s				
16	Ambient Temperature:	22.00 露 Low Pressure Dose: None				Ambient 22.00 露 Low Press: None					Ambient 22.00 露 Low Press: None				
17	Automatic Degas:	Yes				Automatic: Yes					Automatic: Yes				
18															
19															
20															
21															
22															
23															
24															
25	Summary Report					Isotherm Tabular Report					Isotherm Linear Plot				
26															
27															
28															
29	No summary reports could be produced.														
30															
31															
32															

A laser particle size distribution analyzer (BT-2002, Dandong Bettersize Co., Ltd. China) is used to measure the particle size distribution of the microspheres according to the diffraction and scattering phenomenon of the laser, while obtaining the surface

area of the MSs by particle size fitting.

In this work, we mainly prepare polyamine-based microspheres by one-pot method. Next, we prepare to further optimize the preparation process of the aminated microspheres, including reactant feed ratio, reaction temperature, reaction time and the additive amount of glutaraldehyde. In the next adsorption experiment, we will further study the binary and ternary adsorption processes, and the recycling performance of the materials will be studied in detail in the next work.

2. SEM in Fig 4 show those MSs were seriously agglomerated as they been produced. The agglomerated adsorbents present perfect dye remove performance. The conclusion is seemed contradictory.

Author's reply: During the preparation of the aminated microspheres, due to the addition of the polyamine-based substances TETA and PEI, the degree of cross-linking between the molecular chains was increased, causing aggregation. But the amino group content on the same surface area was significantly increased, which in turn led to a higher adsorption amount.

3. Abstract, should contain some important quantitative findings

Author's reply: The comments pointed out above have been revised.

4. Introduction, first paragraph, need to put these references:

Carbohydrate polymers 113, 115-130 (2014).

Journal of hazardous materials 177 (1), 70-80 (2010).

Royal Society Open Science, 4(9), 170697 (2017)

Author's reply: The comments pointed out above have been revised. We have cited these references in the Introduction and first paragraph.

Reviewer #2: Comments to the Author(s)

In the research the authors were studied the preparation of aminated chitosan microspheres by one-pot method and their adsorption properties for dye wastewater. They investigated Characterization of chitosan microspheres with the analysis such as FT-IR, XRD, SEM, TG and EDS, and they also investigated adsorption kinetic constants for chitosan microspheres, thermodynamics parameters and the effects of parameters such as pH on the adsorption process.

Author's reply: We highly appreciate the reviewer's time and constructive suggestions on our manuscript. All the corrections have been made following the reviewer's suggestions and highlighted in BLUE in the revised manuscript.

1. More quantitative measurements be mentioned in the abstract.

Author's reply: The comments pointed out above have been revised.

2. Summary of the section on the conclusion is also presented in the abstract.

Author's reply: The comments pointed out above have been revised. Summary of the section on the conclusion has been presented in the abstract.

3. In the introduction section used more literature. The following references should be cited in this section for upgrading the quality of the manuscript:

3.1 Removal of reactive blue 29 dye by adsorption on modified chitosan in the presence of hydrogen peroxide, Environment Protection Engineering, 42(1), pp. 149-168.

3.2 Synthesis of nanochitosan for the removal of fluoride from aqueous solutions: A study of isotherms, kinetics, and thermodynamics, Fluoride, 50(2), pp. 256-268.

3.3 Synthesis and performance evaluation of chitosan prepared from Persian gulf shrimp shell in removal of reactive blue 29 dye from aqueous solution (Isotherm, thermodynamic and kinetic study). Iranian Journal of Chemistry and Chemical Engineering, 36(3), pp. 25-36.

3.4 Optimization of humic acid removal by adsorption onto bentonite and montmorillonite nanoparticles. Journal of Molecular Liquids, 259, pp. 76-81.

3.5 Investigation of equilibrium, kinetics and thermodynamics of extracted chitin from shrimp shell in reactive blue 29 (RB-29) removal from aqueous solutions. *Desalination and Water Treatment*, 70, pp. 355-363.

3.6 Equilibrium and kinetics study of reactive dyes removal from aqueous solutions by bentonite nanoparticles. *Desalination and Water Treatment*, 97, pp. 329-377.

Author's reply: The comments pointed out above have been revised. We have cited these references in the Introduction and first paragraph.

4. For characterization of adsorbents, because the process was adsorption, so the analysis BET must be made to determine the surface area of the particles.

Author's reply: The surface area of the MSs was characterized by N₂ adsorption/desorption isotherms at 77 K on Accelerated Surface Area and Porosimetry System (ASAP 2020M, Micromeritics Company), and the surface area obtained by the test was almost absent. This may be due to the tight entanglement between the molecular chains of chitosan during the process of crosslinking and precipitation when adding NaOH solution. The BET test data of CTSms, TETA-CTSms and PEI-CTSms microspheres were given.

Row	Column 1 (ASAP 2020 V4.03)	Column 2 (ASAP 2020 Unit 1)	Column 3 (ASAP 2020 Unit 1)
2	ASAP 2020 V4.03 (V4.0) Unit 1	Serial #: 2183	Page 1
5	Sample: MJ-CTSms		Sample: MJ-CTSms
6	Operator: dnb		Operator: dnb
7	Submitter:		Submitter
8	File: D:\2020\DATA\2018\4\MJ-CTSms.SMP		File: D:\2020\DATA\2018\4\MJ-CTSms.SMP
11	Started: 7/28/2018 8	Analysis Adsorptive: N2	Started: 7/28/2018 Analysis N2
12	Completed: 7/28/2018 1	Analysis Bath Temp.: -195.655 °C	Completed: 7/28/2018 Analysis -195.655 °C
13	Report Time: 8/5/2018 1:	Thermal Correction: No	Report T: 8/5/2018 Thermal (No)
14	Sample Mass: 0.1023 g	Warm Free Space: 27.1351 cm ³ /Measure	Sample Ms: 0.1023 g Warm Free: 27.1351 cm ³ /Measure
15	Cold Free Space: 84.5240 cm ³	Equilibration Interval: 10 s	Cold Free: 84.5240 cm ³ Equilibr: 10 s
16	Ambient Temperature: 22.00 °C	Low Pressure Dose: None	Ambient: 22.00 °C Low Press: None
17	Automatic Degas: No		Automatic: No
25	Summary Report		Isotherm Tabular Report
28			Relative Absolute Quantity Elapsed T Saturatic
29	No summary reports could be produced.		0.058579 60.35839 -0.03373 01:15 1030.384 MJ-CTSms - Adsorption
30			0.09063 93.38409 -0.05768 01:17 0.058579 -0.03373 Relative Quantity Adsorbed (mmol/g)
31			0.118435 122.0332 -0.0783 01:20 0.09063 -0.05768
32			0.146352 150.7988 -0.10073 01:22 0.118435 -0.0783

	A	B	C	D	E	F	G	H	I	J	K	L	M	N
1														
2	ASAP 2020 V4.03 (V4.03	Unit 1	Serial #: 2183	Page 1		ASAP 2020	Unit 1	Serial #:	Page 1			ASAP 2020	Unit 1	Serial #:Page
3														
4														
5	Sample:	MJ-PEI-CTSms				Sample:	MJ-PEI-CTSms					Sample:	MJ-PEI-CTSms	
6	Operator:	dhb				Operator:	dhb					Operator:	dhb	
7	Submitter:					Submitter:						Submitter:		
8	File:	D:\2020\DATA\2018\4\MJ-PEI-CTSms.SMP				File:	D:\2020\DATA\2018\4\MJ-PEI-CTSms.SMP					File:	D:\2020\DATA\2018\4\M	
9														
10														
11	Started:	7/28/2018	Analysis Adsorptive:	N2		Started:	7/28/2018	Analysis N2				Started:	7/28/2018	Analysis N2
12	Completed:	7/28/2018	Analysis Bath Temp.:	-195.709	°C	Completed:	7/28/2018	Analysis -195.709	°C			Completed:	7/28/2018	Analysis -19
13	Report Time:	8/5/2018	Thermal Correction:	No		Report Time:	8/5/2018	Thermal CNo				Report Time:	8/5/2018	Thermal CNo
14	Sample Mass:	0.1071 g	Warm Free Space:	27.0121 cm ³	Measure	Sample Mass:	0.1071 g	Warm Free	27.0121 cm ³	Measure		Sample Mass:	0.1071 g	Warm Free
15	Cold Free Space:	82.9712 cm ³	Equilibration Interval:	10 s		Cold Free Space:	82.9712 cm ³	Equilibr:	10 s			Cold Free Space:	82.9712 cm ³	Equilibr:
16	Ambient Temperature:	22.00	°C	Low Pressure Dose:	None	Ambient Temperature:	22.00	°C	Low Press:	None		Ambient Temperature:	22.00	°C
17	Automatic Degas:	Yes				Automatic Degas:	Yes					Automatic Degas:	Yes	
18														
19														
20														
21														
22														
23														
24														
25	Summary Report					Summary Report						Summary Report		
26														
27														
28														
29	No summary reports could be produced.													
30														
31														
32														

	A	B	C	D	E	F	G	H	I	J	K	L	M	N	O
1															
2	ASAP 2020 V4.03 (V4.03	Unit 1	Serial #: 2183	Page 1		ASAP 2020	Unit 1	Serial #:	Page 1			ASAP 2020	Unit 1	Serial #:Page	
3															
4															
5	Sample:	MJ-TETA-CTSms				Sample:	MJ-TETA-CTSms					Sample:	MJ-TETA-CTSms		
6	Operator:	dhb				Operator:	dhb					Operator:	dhb		
7	Submitter:					Submitter:						Submitter:			
8	File:	D:\2020\DATA\2018\4\MJ-TETA-CTSms.SMP				File:	D:\2020\DATA\2018\4\MJ-TETA-CTSms.SMP					File:	D:\2020\DATA\2018\4\MJ-TETA-		
9															
10															
11	Started:	7/29/2018	Analysis Adsorptive:	N2		Started:	7/29/2018	Analysis N2				Started:	7/29/2018	Analysis N2	
12	Completed:	7/29/2018	Analysis Bath Temp.:	-195.710	°C	Completed:	7/29/2018	Analysis -195.710	°C			Completed:	7/29/2018	Analysis -195.71	
13	Report Time:	8/5/2018	Thermal Correction:	No		Report Time:	8/5/2018	Thermal CNo				Report Time:	8/5/2018	Thermal CNo	
14	Sample Mass:	0.0881 g	Warm Free Space:	26.9156 cm ³	Measure	Sample Mass:	0.0881 g	Warm Free	26.9156 cm ³	Measure		Sample Mass:	0.0881 g	Warm Free	
15	Cold Free Space:	83.6708 cm ³	Equilibration Interval:	10 s		Cold Free Space:	83.6708 cm ³	Equilibr:	10 s			Cold Free Space:	83.6708 cm ³	Equilibr:	
16	Ambient Temperature:	22.00	°C	Low Pressure Dose:	None	Ambient Temperature:	22.00	°C	Low Press:	None		Ambient Temperature:	22.00	°C	
17	Automatic Degas:	Yes				Automatic Degas:	Yes					Automatic Degas:	Yes		
18															
19															
20															
21															
22															
23															
24															
25	Summary Report					Summary Report						Summary Report			
26															
27															
28															
29	No summary reports could be produced.														
30															
31															
32															

5. Formulas related to calculation of kinetics and thermodynamics in the section of experimental expressed not in the results section.

Author's reply: The comments pointed out above have been revised.

6. The conclusion is poorly written and must be rewritten.

Author's reply: The comments pointed out above have been revised. The conclusion has been rewritten.

Appendix B

Dear Editor,

We deeply appreciate the time and effort you have spent in reviewing our manuscript (RSOS-182226). Per your instruction, we have included *Ethics statement* alongside the other end statements.

The manuscript has been revised after reading the comments provided by the two reviewers. All the changes have been highlighted in **RED** in the revised manuscript. We would greatly appreciate it if you could approve the publication of the revised manuscript.

Should you have any more questions or advice, please let us know.

Yours sincerely,

Prof. Dr. Jianlan Cui

Reviewer #1: Comments to the Author(s)

Author's reply: We highly appreciate the reviewer's time and constructive suggestions on our manuscript.

1. BET should give data and graphs according to the style in the literature, not screen shots. Ref (1) Chem. Mater. 2001, 13, 3169-3183; (2) Journal of Industrial and Engineering Chemistry, Volume 21, 25 January 2015, Pages 369-377; (3) Colloid Polym Sci (2018) 296:59–70

Author's reply: The surface area obtained by the BET test was almost absent. The BET test data of the microspheres CTSms, TETA-CTSms and PEI-CTSms are given in attached **Excel** files (supplementary materials).

2. My foregoing question of "SEM in Fig 4 show those MSs were seriously agglomerated as they been produced. The agglomerated adsorbents present perfect dye remove performance. The conclusion is seemed contradictory."

Author's reply: During the preparation of the aminated microspheres, due to the addition of the polyamine-based substances TETA and PEI, the degree of cross-linking between the molecular chains was increased, causing aggregation. But the amino group content on the same surface area was significantly increased, which in turn led to a higher adsorption amount.

Thus, please provide evidence data about "the amino group content on the same surface area was significantly increased". This claimed conclusion should be proved by Solid data, not description.

Author's reply: The comments pointed out above have been revised. As shown in **Fig. 5** in the revised manuscript, the *N* content of the microspheres PEI-CTSms and TETA-CTSms increased significantly after the crosslinking reaction.

Fig. 5 EDS for CTSms, PEI-CTSms and TETA-CTSms

Reviewer #2: Comments to the Author(s)

Now the manuscript can be accepted for publication.

Author's reply: We highly appreciate the reviewer's time and constructive suggestions on our manuscript.